# Sleep disorders and risk of infertility: A meta-analysis of observational studies

Xiaoxiao Qin[1]*, Siyun Fang[2], Yaqi Cai[1]

**1** Hospital-Acquired Infection Control Department, Chengdu Women's and Children's Central Hospital, Chengdu City, Sichuan Province, China, **2** Department of English, Chengdu Xichuan Middle School, Chengdu City, Sichuan Province, China

* qinxiaoxiaouk@yeah.net

**Data Availability Statement:** All relevant data are within the manuscript and its Supporting Information files.

**Funding:** Unfunded studies.

## Abstract

### Objective

The purpose of this study was to determine the relationship between sleep disorders and risk of infertility.

### Method

Three databases (PubMed, Embase, and Cochrane Library) were searched form their inception to April 30, 2023. Information of study design, control group and experimental group, number of participants, and study outcomes was extracted. The quality of the studies was evaluated using the Newcastle-Ottawa Scale (NOS scale) and the Agency for Health-care Research and Quality (AHRQ scale). Narrative synthesis and meta-analysis were used to analyze these studies.

### Result

Eight cohort, cross-sectional, and case-control studies were considered. The reviewed studies were high-quality. Pooled analysis showed that the risk of infertility was 1.43-fold higher in patients with sleep disturbance (HR = 1.43, 95% CI, 0.97–2.11, z = 1.79), but this was not statistically different; the risk was 1.58-fold higher in patients with OSA compared to those without OSA (HR = 1.58, 95%, CI, 0.99–2.52, z = 1.91), but this was not statistically significant. Wake-up time is also associated with infertility (OR = 1.14; 95%CI = 1.01–1.28; P = 0.037). For every hour they stay awake beyond 8:00 AM, participants had a 41% higher risk of infertility (P = 0.004). The early-to-bed/late-to-rise (EL), LE, and LL groups had a higher risk of infertility than the EE group.

### Conclusion

The present study did not find an association between sleep disorders and the risk of infertility. Therefore, more observational studies are warranted to explore the association between sleep disorders and the risk of infertility.

**Competing interests:** NO authors have competing interests.

## Introduction

Sleep disorders and infertility are both common global public health issues in contemporary society. Sleep disorders refer to the disruption of normal sleep patterns in the human body [1], while infertility refers to the inability to conceive normally for 12 months or longer [2]. There are many causes of infertility, but recent research suggests that sleep disorders may also be one of the important factors leading to patients developing infertility. The quality of sleep can affect a person's mental state, brain function, metabolism, and hormone levels which in turn affects reproductive function leading to infertility [3]. For example, the likelihood of depression symptoms in people who lack sufficient sleep is ten times higher than those who have adequate sleep, and the likelihood of anxiety symptoms is 17 times higher [4]. At the same time, sleep is closely related to hormones in the body [5]. Lack of sleep can send signals to the body, producing more stress hormones, reducing levels of estrogen, testosterone, and other reproductive hormones, which may affect a person's fertility [6]. In addition, sleeping patterns produce general characteristics that drive necessary hormone synthesis for reproduction physiologically such as secretion and metabolism [7]. Some sex hormones also affect human reproductive function, and their secretion works synergistically with circadian rhythms. Furthermore, good, or bad quality of sleep also affects metabolism, immunity system, hormones, and cardiovascular system [8]. However, the impact of sleep on fertility is often overlooked when studying infertility. There is a significant lack of research on the impact of sleep disorders on reproduction [9].

Currently, most studies focus on sleep problems during pregnancy, particularly in female. And the negative effect of sleep difficulties on sperm quality in male, infertility due to sleep disorders, and the impact on reproductive outcomes have also been reported [10–12]. However, current studies on the association of sleep disorders and risk of infertility have reached mixed conclusions. Therefore, this review systematically examines previous studies [11–18] related to risk of infertility in sleep disorders individuals, including obstructive sleep apnea, with the aim of clarifying whether sleep disorders are associated with infertility risk. This will help to increase public awareness of the problems associated with sleep disorders and improve overall quality of life.

## Material and methods

### Protocol and registration

This review was conducted according to the guidelines of Preferred Reporting Items for Systematic Reviews and Meta-Analyses (PRISMA) [13] and has been registered in PROSPERO with registration number CRD42023399156.

### Search strategy

This review conducted data retrieval from three databases, called PubMed, Embase and Cochrane Library. The keywords used in the search including sleep disorders, infertility, sleep apnea syndrome and other related words if necessary. The search period included studies published before April 30, 2023, and the language was limited to English. If necessary, further search was conducted by using the references and citations of selected papers. In addition, if more information was needed regarding sleep disorders and infertility, latest research will be sought by contacting experts in this field. Individual databases were retrieved independently by individual authors (Qin XX and Fang SY), cross checked against the amount retrieved by the search, and if there was ambiguity was resolved by discussion. For specific search strategies for the three databases is shown in S2 Table.

## Eligibility criteria

Two researchers (Qin XX and Fang SY) independently assessed the eligibility of the studies included in this review. The inclusion criteria for the studies were: 1) observational studies including cohort, cross-sectional, and case-control studies; 2) participants were individuals with or without sleep disorders, with no gender restrictions; 3) sleep disorder diagnosis was based on a sleep disorder scale or clinical diagnosis by a doctor; 4) articles were written in English; 5) did not qualify for study characteristics such as sample size, country.

Conference abstracts, case reports, animal experiments or literature reviews were excluded. The entire inclusion/exclusion process was conducted independently by two researchers. In cases of disagreement, a third reviewer (Cai YQ) should be consulted to minimize heterogeneity among included articles.

## Data extraction

The data in this review was extracted by two researchers (Qin XX and Fang SY). Controversial parts were discussed and reviewed by a third reviewers (Cai YQ). The extracted data included basic information (authors, publication year, country or region, study type, duration of study period, data source, control group and experimental group characteristics such as average age and confounding factors), standards for measuring and diagnosing sleep disorders (doctor diagnosis, health survey questionnaire scores using the International Classification of Diseases, Ninth Revision, Clinical Modification or Medical Diagnostic Index (ICD-9-CM or MDI), basic sample characteristics (number of individuals in control and experimental groups with their respective average ages), and the incidence for each outcome measure by hazard ratio or odds ratio (HR or OR), and 95% confidence interval.

## Quality assessment

The Newcastle Ottawa quality assessment scale (NOS) criteria and AHRQ cross-sectional study evaluation standards [14,15] were selected to assess the quality of the included studies. The AHRQ cross-sectional study evaluation criteria is objective and therefore it is more appropriate to use the AHRQ to evaluate the quality of cross-sectional studies [16]. The highest score on the NOS scale is nine points. Articles with high quality scores receive seven points or more; articles with medium quality scores receive 4 to 4 points; articles with low-quality scores receive less than 3 points. The highest score on AHRQ's cross-sectional study evaluation standard is eleven points: high-quality articles receive six to seven points; articles with medium quality scores receive four to five-and-a-half points; articles with low-quality scores receive less than four points.

## Statistical analysis

Sleep disorders are usually determined by International Classification of Diseases, Ninth Revision, Clinical Modification (ICD-9-CM), MDI Scores, or a doctor's diagnosis. Some articles [19] also define sleep disorders as the effect of different bedtime and wake times on infertility (e.g., going to bed at 10, 11, or 12 p.m., waking up after 5, 6, or 7 a.m.).

Since the included studies provided sufficient data, we were able to conduct a meta-analysis. We quantified the risk of infertility associated with sleep disorders using combined OR or HR and 95% CI. The heterogeneity between different studies was determined by $I^2$ and P values from chi-square tests. The range of $I^2$ was from 0% to 100%, and a random-effects model was used for this meta-analysis. When sufficient data were available, further analysis was conducted on different types of sleep disorders. Subgroup analysis based on follow-up period of

the studies was also performed to examine the source of heterogeneity. Sensitivity analysis was conducted to test the influence of individual studies on the combined data as well as their quality. In addition, publication bias testing was performed using funnel plots and Egger's test. All statistical analyses were carried out using Stata 14, and Review Manager 5.4.

## Results

### Study screening

The search of three databases (PubMed, Cochrane and Embase) resulted in a final search of 1339 articles. After reading the full text, 33 articles were excluded for the following reasons. Among them, infinity leading to sleep disorders (n = 13), only meeting abstract (n = 4), and the type of study does not match (n = 16). Eight articles [17–24] were finally included, one of which was searched from the relevant articles. The included studies were all observational studies, excluding other types of studies. Three of the studies were cohort studies, [17, 18, 23] three studies were case-control studies [19–21] and two studies were cross-sectional studies [20, 24]. The detailed selection process was shown in Fig 1. PRISMA diagram. In addition, the basic characteristics included in the literature are shown in Table 1.

### Study quality

The NOS scale showed that the 6 studies included in this review [17–19, 21, 22, 24] ranged from 6 to 9 points. Among them, there were 5 high-quality studies [17–19, 21, 22] (83.3%) and one medium-quality study (18) (0.17%). The AHRQ scale showed that the two articles

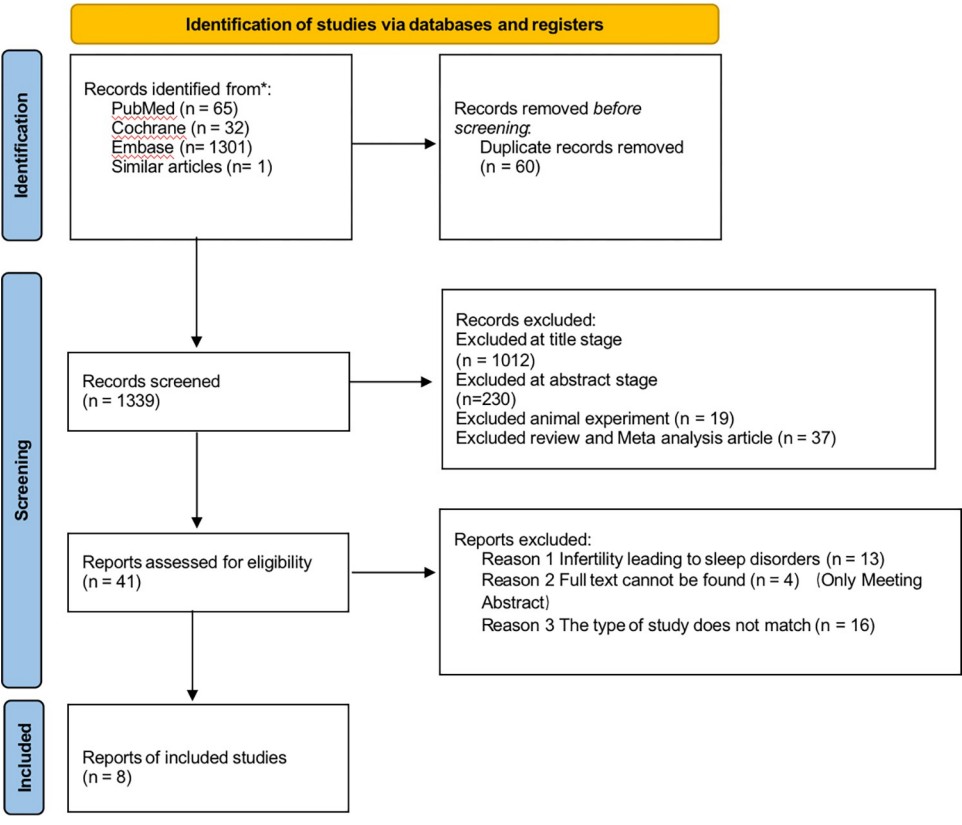

**Fig 1. PRISMA diagram.**

**Table 1. The characteristics of studies.**

| Author | Year | Country | Study Type | Follow-up Years | Diagnostic criteria | Case group | Control group | Age (Mean ± SD) | Confounders adjusted |
|---|---|---|---|---|---|---|---|---|---|
| I-Duo Wang | 2018 | China | Cohort Study | 10 years | ICD-9-CM | 16,718 | 33,436 | Case group: 42.46 ± 7.06 Control group: 43.22 ± 7.51 | Age, Hypertension, DM, Hyperlipidemia, COPD, CKD, IHD, HF, Stroke, Cancer, Obesity, Hypoestrogenism, Polycystic ovaries, Irregular menstrual cycle, Endometriosis, Uterine leiomyoma, Cushing's syndrome, Thyrotoxicosis with or without goiter, Acquired hypothyroidism, Anxiety, Depression, Urbanization level, Insured premium |
| Lauren Anne Wise | 2018 | United States and Canada. | Cohort Study | 4 years | Men during the preconception period and aged ≥21 years | 1176 | Normal male | 31 | Physician diagnosis and MDI score |
| Sydney Kaye Willis | 2019 | North American | Cohort Study | 5 years | Female participants aged 21–45 years attempting pregnancy | 6873 | Normal Female | 29.9 | Physician diagnosis and MDI score |
| Jinyan Zhao | 2023 | US | Cross-sectional Study | 5 years | the question RHQ074 (the reproductive health questionnaire) | 248 | 1572 | Case group: 31.8 Control group: 31.7 | |
| Pin-Yao Lin | 2022 | China | the Nested Case-Control Study | 16 years | ICD-9-CM | 1607 | 3214 | none | OSA, Urbanization, Length of hospital stays, Overweight and obesity, Hyperlipidemia, Asthma, COPD, Chronic liver diseases, Chronic kidney diseases, Major psychiatric disorders, Minor psychiatric disorders |
| Zhu Wei Lim | 2021 | China | Case-Control Study | 13 years | ICD-9-CM | 1946 | 3892 | Case group: 32.19 ± 6.20 Control group: 32.24 ± 6.37 | Insured premium, IHD, Cancer, Obesity, Hyperestrogenism, Polycystic ovaries, Irregular menstrual cycle, Endometriosis, Uterine leiomyoma, Anxiety, Depression, Season, Urbanization level, Level of care |
| Zhu Liang | 2022 | US | Cross-sectional Study | 3 years | the question RHQ074 (the reproductive health questionnaire). | 212 | 1963 | Case group: 34.13 (6.69) Control group: 30.39 (8.06) | Bedtime, Waketime, Sleep duration (hours), Age, BMI, Poverty level index, Physical activity, Education, Race, Marital status Cotinine |
| Yi-Han Jhuang | 2021 | China | Case-control population-based study | 13 years | ICD-9-CM | 4607 | 18428 | Case group: 34.18 (5.44) Control group: 34.28 (5.81) | Age, Hypertension, Diabetes, Hyperlipidemia, COPD, CKD, CAD, CHF, Liver cirrhosis, Stroke, Cancer, Obesity, Epilepsy, HBV, HCV, Anxiety, Depression |

**Notes:** Abbreviations: HR, Hazard ratio; RM, Recurrent miscarriage group; RIF, Recurrent implantation failure group; OSA, Obstructive sleep apnea; DM, Diabetes mellitus; COPD, Chronic obstructive pulmonary disease; CKD, Chronic kidney disease; IHD, Ischemic heart disease; HF, Heart failure; MDI, the Major Depression Inventory; CAD, Coronary artery disease; CHF, Congestive heart failure; HBV, Hepatitis B virus; HCV, Hepatitis C virus.

included [19, 20] scored 6 and 7 points respectively and were both high-quality studies. See S1 Table for detailed information.

## Systematic review and meta-analysis

The results of the research included in this review were combined with a random effects model to determine whether there is a correlation between sleep disorders and infertility risk. Of the final eight studies, seven [17, 18, 21–24] were included in the meta-analysis. As the results of

the remaining study [19], which investigated the relationship between sleep disorders and infertility based on wake-up time and bedtime, differed from the other studies, it could not be classified into the same category for meta-analysis.

The seven studies included in the meta-analysis were divided into two groups based on the categories of their respective study results. The first meta-analysis was conducted on five studies [17, 18, 21–24] that were grouped based on whether participants had sleep disorders or not. The results of meta-analysis the first group showed that after selecting random effects for merging, HR = 1.43 (95%CI, 0.97–2.11), Z = 1.79. The conclusion drawn from the meta-analysis is that patients with sleep disorders have a 1.43 times higher chance of suffering from infertility compared to those without sleep disorders. Although the results of the meta-analysis are significant, there is no statistical significance in the difference between the experimental group with sleep disorders and the control group without sleep disorders. However, there was high heterogeneity in the meta-analysis $I^2$ = 93%, considering that each study had a different follow-up period. Therefore, heterogeneity analysis was performed on these five studies [17, 18, 21–24]. The result showed P = 0.736>0 .05, which indicated that follow-up periods did not significantly affect effect size. In addition, sensitivity analysis also indicated that none of these five studies [17, 18, 21–24] had a significant impact on the results. Thus, the findings of these studies are relatively stable.

Another meta-analysis was conducted based on three studies [21, 22, 24] with the research outcome of OSA without surgery. OSA is also a type of sleep disorder disease. Patients with OSA often suffer from sleep difficulties. The results of the meta-analysis showed that the combined HR was 1.58 (95% CI, 0.99–2.52), Z = 1.91. From the results of the analysis, patients with OSA had a 1.58 times higher risk of infertility compared to those without OSA. The result of this group is the same as the first group, in that there is still no statistically significant difference between the experimental and observation groups in this group. This group's meta-analysis had moderate heterogeneity $I^2$ = 57%>50%, so it is necessary to explore the reasons for this heterogeneity in this group. Similar to the first group, each study included in this group had different follow-up periods. Therefore, an analysis of heterogeneity related to follow-up period was performed and resulted in P = 0.283>0 .05, which meant that follow-up period may not affect effect size. The sensitivity analysis revealed that the results obtained from random-effect analyses from these three articles [21, 22, 24] were stable and reliable. From the results of the meta-analyses of the above two groups, there were high levels of heterogeneity in both analyses which could possibly be due to all studies included being observational studies. The detailed of the meta-analysis results can be found in Fig 2. The impact of evaluating sleep disorders on infertility and Fig 3. Evaluation of the impact of OSA on infertility.

## Publication bias

No publication bias was found between studies of different quality levels. The bias test for combining the results of the two groups of studies showed that Egger's test indicated

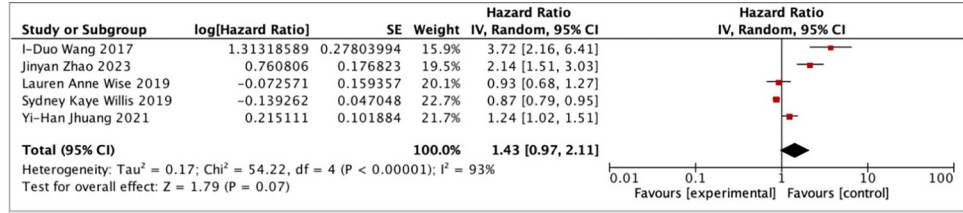

**Fig 2. The impact of evaluating sleep disorders on infertility.**

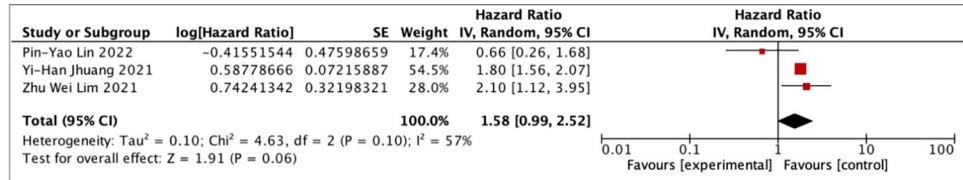

**Fig 3. Evaluation of the impact of OSA on infertility.**

P = 0.057>0.05 for group one [17–19, 20, 24], indicating no publication bias among the five analyzed studies; group two [21, 22, 24] had P = 0.617 >0.05, which was also with no publication bias among these three analyzed studies.

## Discussion

### Main finding

This review summarizes the results of research on the relationship between sleep disorders and infertility over the past few decades. The follow-up periods for the eight studies included in this analysis ranged from 3 to 16 years, making the results reliable [17–24]. Only one study [19] was not included in meta-analysis, while the other seven were grouped and analyzed separately. Although the results of the meta-analysis did not show statistical significance, this may be due to variations in the definitions of sleep disorders and infertility among different studies or other biases that could have affected the results. However, it can be seen from the results that people with sleep disorders have a significantly higher risk of infertility than those without. The findings of this review are consistent with those of previous studies. For example, Ou Zhong *et al.* [25] found that sleep disorders have a negative impact on male reproductive health by affecting sperm quality and leading to fertility problems. In addition, another animal experiment by Jung-Liang Wu [26] on the effects of sleep deprivation on serum testosterone levels in rats also demonstrated that good or bad quality of sleep can affect male rat testosterone levels, which may lead to fertility problems. These studies all indicate a close relationship between sleep and fertility. Furthermore, a study [19] that was not included in the analysis found a significant correlation between bedtime and infertility (OR = 1.24; 95%CI = 1.10–1.40; P<0.001). Similarly, wake-up time was also significantly correlated with infertility (OR = 1.14; 95% CI = 1.01–1.28; P = 0.037). Specifically, there was no significant correlation between different wake-up times and infertility when waking up between midnight and 8:00 am. However, there was a turning point at 8:00 am were waking up later increased infertility risk by 41% per hour (P = 0 .004). This study also divided participants into four groups based on their sleeping habits: early to bed/early to rise (EE), early to bed/late to rise (EL), late to bed/early to rise (LE), and late to bed/late to rise (LL). Participants in EE group had lowest probability of being infertile compared with those in EL, LE or LL groups who had higher probabilities of being infertile.

What's more, Pin-Yao Lin *et al.* [21] found that male patients with OSA were more likely to suffer from infertility problems compared with female patients with OSA. The surgical treatment for men increases their risk of becoming infertile as well. Patients suffering from OSA disease were more likely to be males than females, as found in the studies of Zhu Wei Lim et al. [22] and Yi-Han Jhuang *et al.* [24] This result was consistent with the research results of Young T *et al.* [27] and Fietze I *et al.*, [28] which shows that OSA has a greater impact on men. Age also affects infertility risk for patients who have difficulty sleeping. For those who only suffer from sleep disorders, younger age groups are less likely to experience infertility problems, while patients over 40 years old are positively correlated with infertility [17, 19, 20]. However, according to I-Duo Wang's study [17], women aged 20–25 suffering from NASD-induced

sleep disorders were more likely to become infertile than those aged ≥41. This was opposite to the results for patients with simple sleep difficulties.

In clinical practice, there are many causes of infertility. Obesity, alcoholism, and smoking are some examples, but less attention is paid to infertility caused directly by sleep disorders. According to the previous studies, different sleep disorders can lead to different risks of infertility depending on age and gender. It is necessary to develop corresponding intervention measures based on the actual situation of patients in order to better reduce their risk of infertility. We cannot simply focus on whether patients have sleep disorders; we also need to consider the causes of these disorders as well as their age and gender when developing a comprehensive treatment plan. This is an important aspect that clinical workers need to pay attention to urgently.

## Strengths and limitations

This review summarizes several observational studies on the relationship between sleep disorders and risk of infertility and includes a comprehensive range of studies. The research included in this article was conducted not only in developed regions but also in developing countries. Additionally, the follow-up periods for the included studies were relatively long, ranging from 3 to 16 years, which make the results reliable. In addition, according to the results of the NOS scale and AHRQ scale, there are no low-quality articles included in this review; therefore, the results of this review are valuable as a reference.

However, it is important to consider some limitations of this article and our meta-analysis. Firstly, we only searched three databases (PubMed, Cochrane and Embase) with language limited to English, which may have resulted in incomplete search coverage. Secondly, due to limited relevant studies available at present, there were only eight articles that met inclusion criteria for our analysis which is a small number compared with other reviews or meta-analyses on related topics. In addition, although both developed countries and developing countries are represented among these studies most come from China or USA, there is still room for more diverse geographic representation. Finally, the heterogeneity of text Meta-analysis was high possibly because all participating studies were observational ones; differences in study design, population characteristics, and outcome measurement could be contributing factors leading to higher heterogeneity. Moreover, when conducting Meta-analysis, we combined cross-sectional studies, cohort studies, and case-control studies together, which might also contribute to higher heterogeneity.

## Conclusion

This study shows no statistical difference in the association of sleep disorders with the risk of infertility. In modern societies with an accelerating pace of work and life, more and more people are prone to sleep disorders for various reasons, affecting physical health. However, there is limited evidence on the association between sleep disorders and risk of infertility, and little is known about how sleep disorders affect fertility. Therefore, the results of this study can provide some research ideas and directions for future research. It may also increase public awareness of their own health, such as sleep quality, duration, waking time, and respiratory diseases that may affect sleep. This will allow them to seek professional treatment as early as possible to avoid further deterioration of their health.

## Supporting information

**S1 Checklist.**
(DOCX)

**S1 Table. The NOS scale and AHRQ scale for studies.**
(DOCX)

**S2 Table. Search strategy.**
(DOCX)

## Acknowledgments

We would like to acknowledge the reviewers for their helpful comments on this paper.

## Author Contributions

**Conceptualization:** Xiaoxiao Qin.

**Data curation:** Xiaoxiao Qin.

**Formal analysis:** Xiaoxiao Qin.

**Investigation:** Yaqi Cai.

**Methodology:** Xiaoxiao Qin, Siyun Fang, Yaqi Cai.

**Project administration:** Xiaoxiao Qin, Yaqi Cai.

**Resources:** Siyun Fang.

**Software:** Xiaoxiao Qin.

**Supervision:** Xiaoxiao Qin.

**Validation:** Xiaoxiao Qin, Siyun Fang.

**Writing – original draft:** Xiaoxiao Qin.

**Writing – review & editing:** Xiaoxiao Qin, Siyun Fang, Yaqi Cai.

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
