## [Decision Letter · Decision Letter 0]

27 Sep 2023

PONE-D-23-18697Sleep disorders and risk of infertility: A meta-analysis of observational studiesPLOS ONE

Dear Dr. Qin,

Thank you for submitting your manuscript to PLOS ONE. After careful consideration, we feel that it has merit but does not fully meet PLOS ONE’s publication criteria as it currently stands. Therefore, we invite you to submit a revised version of the manuscript that addresses the points raised during the review process.

Major Revision

We look forward to receiving your revised manuscript.

Kind regards,

Omid Dadras, MD, PhD

Academic Editor

PLOS ONE

3. Please amend your authorship list in your manuscript file to include authors Xiaoxiao Qin, Siyun Fang and Yaqi Cai.

5. We notice that your supplementary tables are included in the manuscript file. Please remove them and upload them with the file type 'Supporting Information'. Please ensure that each Supporting Information file has a legend listed in the manuscript after the references list.

Reviewers' comments:

Reviewer's Responses to Questions

**Comments to the Author**

1. Is the manuscript technically sound, and do the data support the conclusions?

Reviewer #1: No

Reviewer #2: Yes

Reviewer #3: Yes

2. Has the statistical analysis been performed appropriately and rigorously? 

Reviewer #1: I Don't Know

Reviewer #2: Yes

Reviewer #3: Yes

3. Have the authors made all data underlying the findings in their manuscript fully available?

Reviewer #1: Yes

Reviewer #2: Yes

Reviewer #3: Yes

4. Is the manuscript presented in an intelligible fashion and written in standard English?

Reviewer #1: Yes

Reviewer #2: Yes

Reviewer #3: Yes

5. Review Comments to the Author

Reviewer #1: Dear Authors

The manuscript is about sleep disorders and risk of infertility.

The abstract is not understandable without reading text.

Although the included study were small but The manuscript did not show significant finding.

The topic is about sleep disorders but in study most important finding were early to bed ,early to wake association with infertility.

Reviewer #2: The authors performed a systematic review and meta-analysis to determine the association between sleep disturbances and infertility risk, and ultimately found no association between sleep disturbances and infertility risk.

An original, very well written article and a very interesting topic. I suggest that the article be accepted.

Reviewer #3: Comments of Reviewer 2:

1. This is already a well written and properly done meta-analysis.

2. In the introduction, the authors explicitly describe the objective of the study and focused study question included.

3. In the introduction, they describe more about the effect of sleep problem/ disorders on infertility not convincingly demonstrated in studies.

4. Comprehensive literature search conducted and sources was listed (ie, PubMed, Cochrane database) and terms used for electronic literature search provided

5. Please note the reasonable limitations placed on search for example language or ext.

6. Is there any manual search performed and what is authors’ strategy about unpublished data?

7. Are there any disagreements between authors and how they were resolved it?

8. Please characteristics of studies listed (sample size, patient demographics….) and also inclusion and exclusion criteria

9. Please need to analysis or evaluate related to sleep problem and sleep disorder?

10. How many excluded studies are there? Please mention that the reasons for exclusion?

11. Appropriate statistical methods were used to combine results. But please describe the sensitivity analysis was conducted in the study

6. PLOS authors have the option to publish the peer review history of their article (what does this mean?). If published, this will include your full peer review and any attached files.

Reviewer #1: **Yes: **babak amra

Reviewer #2: **Yes: **Bilgehan Atilgan Acar

Reviewer #3: No

---

## [Author Response · Author response to Decision Letter 0]

10 Oct 2023

Dear reviewer 1:

On behalf of my co-authors, we thank you very much for your comments on our manuscript entitled “Sleep disorders and risk of infertility: A meta-analysis of observational studies”. (Manuscript ID: PONE-D-23-18697)

We appreciate for your constructive and valuable comments. We have revised our manuscript considerably according to your comments, questions, and suggestions. In the event that we missed any one of the comments please let us know. This document includes our responses to your comments point by point, and the revised portion are marked in Red in revised manuscript. 

Comment 1: The manuscript is about sleep disorders and risk of infertility. The abstract is not understandable without reading text.

Reply 1: Thank you for your pointed and constructive comments. We have re-adapted and revised the language and expression of the abstract for the objective, methods, results, conclusions and look forward to meeting your requirements.

Changes in revised manuscript: 

Page 1~2, Line 9~33 in red.

Objective

The purpose of this study was to determine the relationship between sleep disorders and risk of infertility. 

Method

Three databases (PubMed, Embase, and Cochrane Library) were searched form their inception to April 30, 2023. Information of study design, control group and experimental group, number of participants, and study outcomes was extracted. The quality of the studies was evaluated using the Newcastle-Ottawa Scale (NOS scale) and the Agency for Healthcare Research and Quality (AHRQ scale). Narrative synthesis and meta-analysis were used to analyze these studies. 

Result

Eight cohort, cross-sectional, and case-control studies were considered. The reviewed studies were high-quality. Pooled analysis showed that the risk of infertility was 1.43-fold higher in patients with sleep disturbance (HR = 1.43, 95% CI, 0.97-2.11, z = 1.79), but this was not statistically different; the risk was 1.58-fold higher in patients with OSA compared to those without OSA (HR = 1.58, 95%, CI, 0.99-2.52, z = 1.91), but this was not statistically significant. Wake-up time is also associated with infertility (OR=1.14; 95%CI=1.01-1.28; P=0.037). For every hour they stay awake beyond 8:00 AM, participants had a 41% higher risk of infertility (P=0.004). The early-to-bed/late-to-rise (EL), LE, and LL groups had a higher risk of infertility than the EE group. 

Conclusion

The present study did not find an association between sleep disorders and the risk of infertility. Therefore, more observational studies are warranted to explore the association between sleep disorders and the risk of infertility.

Comment 2: Although the included study were small but the manuscript did not show significant finding.

Reply 2: Thank you for your pointed comments. Although we found that sleep disturbance was not associated with the risk of infertility, the conclusion we reached through a systematic and comprehensive search of the literature, collection and extraction of information, rigorous quality appraisal, and production forest plots of science is believed to be the current evidence for more reliable observational studies. Furthermore, we emphasize in our conclusions that more observational studies are needed in the future to explore the association between them.

Changes in revised manuscript: 

Page 2, Line 31~33 in red.

The present study did not find an association between sleep disorders and the risk of infertility. Therefore, more observational studies are warranted to explore the association between sleep disorders and the risk of infertility.

Comment 3: The topic is about sleep disorders but in study most important finding were early to bed, early to wake association with infertility.

Reply 3: Thank you for your reminder. We also found that the association between early sleeping and infertility risk varied among different studies before deciding to conduct this meta-analysis work. Unfortunately, the results of the meta-analysis do not support the above notion. Therefore, we need a prudent view of the conclusions of these studies. Your support is also awaited.

Under your kind help and professional guidance, we believe that our manuscript has been improved substantially. We are looking forward to your further suggestions.

Sincerely yours,

Xiaoxiao Qin

E-mail:qinxiaoxiaouk@yeah.net

Dear reviewer 2:

On behalf of my co-authors, we thank you very much for your comments on our manuscript entitled “Sleep disorders and risk of infertility: A meta-analysis of observational studies”. (Manuscript ID: PONE-D-23-18697)

We appreciate for your constructive and valuable comments. We have revised our manuscript considerably according to your comments, questions, and suggestions. In the event that we missed any one of the comments please let us know. This document includes our responses to your comments point by point, and the revised portion are marked in Red in revised manuscript. 

Comment 1: The authors performed a systematic review and meta-analysis to determine the association between sleep disturbances and infertility risk, and ultimately found no association between sleep disturbances and infertility risk. An original, very well written article and a very interesting topic. I suggest that the article be accepted.

Reply 1: Thank you for your acknowledgement and encouragement to our manuscript, it is your encouragement that encourages us to make constant progress and move forward. In addition, we appreciate your constructive suggestions and comments, and we have made point-to-point revisions in the hope that the quality of our manuscript will be greatly enhanced under your guidance.

Comment 2: This is already a well written and properly done meta-analysis.

Reply 2: Thank you for your acknowledgement and encouragement to our manuscript, it is your encouragement that encourages us to make constant progress and move forward.

Comment 3: In the introduction, they describe more about the effect of sleep problem/ disorders on infertility not convincingly demonstrated in studies.

Reply 3: Thank you for your friendly concerns. We note that most current studies have focused on sleep problems during pregnancy, particularly in women. The negative impact of sleep difficulties on male sperm quality, infertility due to sleep disturbances, and the impact on reproductive outcomes have also been reported. However, current studies on the association between sleep disturbances and the risk of infertility have reached mixed conclusions. So, we just conducted this systematic review and meta-analysis. These we elaborate upon in the Introduction, hoping to meet your expectations. 

Changes in revised manuscript: 

Page 3~4, Line 56~65 in red.

Currently, most studies focus on sleep problems during pregnancy, particularly in female. And the negative effect of sleep difficulties on sperm quality in male, infertility due to sleep disorders, and the impact on reproductive outcomes have also been reported [10-12]. However, current studies on the association of sleep disorders and risk of infertility have reached mixed conclusions. Therefore, this review systematically examines previous studies [11-18] related to risk of infertility in sleep disorders individuals, including obstructive sleep apnea, with the aim of clarifying whether sleep disorders are associated with infertility risk. This will help to increase public awareness of the problems associated with sleep disorders and improve overall quality of life.

Comment 4: Comprehensive literature search conducted and sources was listed (ie, PubMed, Cochrane database) and terms used for electronic literature search provided.

Reply 4: Thanks for your affirmation, we present the retrieval strategy and retrieval steps in detail in the Surporting Information (S2 Table. Search Strategy) and also facilitate readers for further review and reading.

Comment 5: Please note the reasonable limitations placed on search for example language or ext.

Reply 5: Thanks for your friendly reminder, we have included a description of the language restrictions in our search strategy.

Changes in revised manuscript: 

Page 4, Line 74~76 in red.

The search period included studies published before April 30, 2023, and the language was limited to English.

Comment 6: Is there any manual search performed and what is authors’ strategy about unpublished data?

Reply 6: Thank you for your kind reminder, we preferentially retrieved the literature included in our database while also reviewing the references of the included studies, avoiding missing possible grey or unpublished literature.

Changes in revised manuscript: 

Page 4, Line 76~79 in red.

If necessary, further search was conducted by using the references and citations of selected papers. In addition, if more information was needed regarding sleep disorders and infertility, latest research will be sought by contacting experts in this field

Comment 7: Are there any disagreements between authors and how they were resolved it?

Reply 7: Thank you for your friendly reminder. We retrieved data in 2 authors independently, cross checking the number retrieved, and if there was ambiguity was resolved by discussion.

Changes in revised manuscript: 

Page 4, Line 79~81 in red.

Individual databases were retrieved independently by individual authors (Qin XX and Fang SY), cross checked against the amount retrieved by the search, and if there was ambiguity was resolved by discussion. 

Comment 8: Please characteristics of studies listed (sample size, patient demographics….) and also inclusion and exclusion criteria.

Reply 8: Thank you for your friendly and constructive reminder. We added a statement to the inclusion criteria not restricting the sample size, country and skin colour of included studies.

Changes in revised manuscript: 

Page 5, Line 89~90 in red.

5）did not qualify for study characteristics such as sample size, country. 

Comment 9: Please need to analysis or evaluate related to sleep problem and sleep disorder?

Reply 9: Thank you for your concern. The different sleep disorders distinguished in our manuscript, containing poor sleep quality, sleep apnea, and various sleep habits, among others, have been systematically reviewed and subgroup analyses in the manuscript. Therefore, it does not feel necessary to make the next distinction, and you look forward to your next suggestions.

Comment 10: How many excluded studies are there? Please mention that the reasons for exclusion?

Reply 10: Thank you for your friendly reminder, we have increased the number of excluded literatures and the reason for exclusion in our literature screening.

Changes in revised manuscript: 

Page 7, Line 139~141 in red.

After reading the full text, 33 articles were excluded for the following reasons. Among them, infinity leading to sleep disorders (n = 13), only meeting abstract (n = 4), and the type of study does not match (n = 16). 

Comment 11: Appropriate statistical methods were used to combine results. But please describe the sensitivity analysis was conducted in the study.

Reply 11: Thank you for your concern, we have methods and results about sensitivity analysis in the manuscript.

Changes in revised manuscript: 

Page 7, Line 132~133 in red.

Sensitivity analysis was conducted to test the influence of individual studies on the combined data as well as their quality. 

Page 10, Line 190~192 in red.

The sensitivity analysis revealed that the results obtained from random-effect analyses from these three articles [21, 22, 24] were stable and reliable.

We really appreciate your positive and insightful suggestions on our manuscript. Under your kind help and professional guidance, we believe that our manuscript has been improved substantially. We are looking forward to your further suggestions.

Sincerely yours,

Xiaoxiao Qin

E-mail:qinxiaoxiaouk@yeah.net

---

## [Editor Report · Decision Letter 1]

16 Oct 2023

Sleep disorders and risk of infertility: A meta-analysis of observational studies

PONE-D-23-18697R1

Dear Dr. XIAOXIAO QIN,

We’re pleased to inform you that your manuscript has been judged scientifically suitable for publication and will be formally accepted for publication once it meets all outstanding technical requirements.

Kind regards,

Omid Dadras, MD, PhD

Academic Editor

PLOS ONE
---

## [Editor Report · Acceptance letter]

20 Oct 2023

PONE-D-23-18697R1 

Sleep disorders and risk of infertility: A meta-analysis of observational studies 

Dear Dr. Qin:

I'm pleased to inform you that your manuscript has been deemed suitable for publication in PLOS ONE. Congratulations! Your manuscript is now with our production department. 

Kind regards, 

on behalf of

Dr Omid Dadras 

Academic Editor

PLOS ONE